# Nanomaterials Used in Conservation and Restoration of Cultural Heritage: An Up-to-Date Overview

**DOI:** 10.3390/ma13092064

**Published:** 2020-04-29

**Authors:** Madalina Elena David, Rodica-Mariana Ion, Ramona Marina Grigorescu, Lorena Iancu, Elena Ramona Andrei

**Affiliations:** 1“Evaluation and Conservation of Cultural Heritage” Research Group, National Institute for Research and Development in Chemistry and Petrochemistry–ICECHIM, 060021 Bucharest, Romania; madalina.e.david@gmail.com (M.E.D.); rmgrigorescu@gmail.com (R.M.G.); lorenna77ro@yahoo.com (L.I.); andreiramona@hotmail.com (E.R.A.); 2Doctoral School of Materials Engineering Department, Valahia University, 130104 Targoviste, Romania

**Keywords:** carbon nanotubes, nanomaterials, metal nanoparticles, hydroxyapatites, cultural heritage, carbonated hydroxyapatite

## Abstract

In the last few years, the preservation of cultural heritage has become an important issue globally, due to the fact that artifacts and monuments are continually threatened by degradation. It is thus very important to find adequate consolidators that are capable of saving and maintaining the natural aspect of these objects. This study aims to provide an updated survey of the main nanomaterials used for the conservation and restoration of cultural heritage. In the last few years, besides the classic nanomaterials used in this field, such as metal nanoparticles (copper and silver) and metal oxides (zinc and aluminum), hydroxyapatite and carbonated derivatives, tubular nanomaterials (such as carbon nanotubes) have been used as a potential consolidate material of cultural heritage. Tubular nanomaterials have attracted attention for use in different fields due to their structures, as well as their ability to present multiple walls. These nanotubes have the necessary properties in preserving cultural heritage, such as superior mechanical and elastic strength (even higher than steel), high hydrophobicity (with a contact angle up to 140°), optical properties (high photodegradation protection), large specific surface area (from 50 to 1315 m^2^/g, depending on the number of walls) for absorption of other nanomaterials and relatively good biocompatibility.

## 1. Introduction

The preservation of cultural heritage is essential for humanity to maintain the history of mankind, as well as the authenticity of artifacts and constructions. An artifact represents any object created or modified by humans bearing historical value. In archeology, an artifact is an object recovered by archaeological methods which may have a cultural interest. These artifacts are continually threatened by degradation factors. For example, stone, paper and wood artifacts are constantly subject to several serious degradation factors, such as biological or chemical degradation, which affect more or less the structural integrity and mechanical strength of these materials [1,2,3,4]. Nanomaterials (1–100 nm) with higher surface areas than similar larger-scale materials have the possibility to penetrate deep into the damaged artifacts due to their particle size [5].

In the last few decades, nanomaterials have received special attention in the field of cultural heritage due to their unique properties. Currently, the obtaining of new systems (such as metal oxides and hydroxides (TiO_2_, ZnO, Ca(OH)_2_, Mg(OH)_2_, Sr(OH)_2_, and metal nanoparticles (Au, Ag, Pt)) and their potential as consolidants on different artifacts and works of art have been reported in the literature [1,6,7]. For example, metal oxide nanoparticles have been used in the last decade to protect building surfaces against biofilm formation. The potential of these nanoparticles in the conservation and restoration of cultural heritage has been established for the consolidation of decomposed materials, self-cleaning, improving the surface of the material or as biocide to reduce biodeterioration [8].

Presently, the efficiency of these nanoparticles has been tested on different materials belonging to the cultural heritage. Barberio M. and co-workers investigated the possibility of using nanoparticles of TiO_2_ and SiO_2_ as consolidating materials without introducing chemical, physical or aesthetic changes on the surfaces of ceramic artifacts. After the artifacts were treated, it was observed that this layer was perfectly transparent, uniform and hydrophobic, and the nanoparticles penetrated the surface of the artifacts, giving them a higher resistance [9]. In the case of wooden artifacts, several studies that evaluate the performance of Ag, Cu, ZnO, and TiO_2_ nanoparticles have been reported. Tests against termites, rot, mold, fungi and UV degradation have shown that these nanoparticles significantly improve the wood’s resistance and provide protection against degradation. Also, it has been reported that ZnO and TiO_2_ nanoparticles have promising antifungal and antibacterial properties [10,11,12]. Ciliberto E. and co-workers obtained Sr(OH)_2_ nanoparticles and applied them on various cultural heritage artifacts (wood, paintings, paper and stone). Following the experiments, they showed that these nanoparticles can be successfully used for the protection and consolidation of cultural heritage artifacts [13]. In another study, MgO nanoparticles were successfully used to de-acidify the paper, ensuring the prevention of paper degradation [14,15]. Sassoni E. and co-workers and Ion R.-M. and co-workers were pioneers in the use of hydroxyapatite (HAp) as an alternative to calcium oxalate for the consolidation of carbonate stones used in building heritage [16,17]. Also, based on its good compatibility with the crystal structure and lattice parameters of calcite, HAp has been applied for the consolidation of limestones [16], marbles [18] and chalk stone [19,20]. Thanks to its low viscosity, this aqueous consolidant product is able to penetrate deeply into the stone, generating significant improvement in mechanical properties [21]. Presently, carbonated hydroxyapatite and its metallic derivatives seem to be alternatives for older consolidants [22].

Referring to all nanoparticles used to consolidate different artifacts, the situation of the diffusion over time of papers related to the application of nanoparticles on different artifacts is presented in Figure 1.

In recent years, researchers have tried to find new materials in order to improve the properties of consolidating materials (Figure 2). From this point of view, nanotubes have received a real interest from researchers because of their ideal properties: open interior and a large volume (reported to the size of the tube), which makes the inner surface accessible and thus allows the attachment of different nanomaterials/agents inside the tubes; the surface porosity or its shape does not change with pH variations; not being vulnerable to microbial attack; high mechanical and elasticity resistance [23,24].

Currently, several studies have been reported regarding the successful use of tubular materials in art conservation [25]. Also, a new challenge regarding these tubular materials is to attach nanoparticles on the surface of nanotubes in order to test their potential in preserving/restoring cultural heritage [26,27].

## 2. Types of Consolidating Materials and Substrates

Wood and stone are the oldest materials used in the construction of different types of structure founded in many parts of the world [28]. Initially, these materials were used in the construction of different buildings (houses, churches, etc.), and later they were used to make various decorative objects (furniture, statues, instruments, etc.) [29]. One of the most important applications of wood, which has been discovered by humans, remains the manufacture of paper [3].

Wood is an organic material that is continuously subject to damage through various processes encountered in nature (heat, frost, the presence of various organisms, etc.). The most used wood in the manufacture of artifacts is poplar, lime and spruce [30,31]. In the case of the wood, several factors that affect its integrity have been identified:

extreme temperatures, that lead to the loss of the structural strength of the wood [1];

biological attack (insects and fungi) which makes the wood becomes soft and fragile [32,33];

relative high humidity, that can cause severe alteration of the wood substrate [30].

Thus, the consolidation of fragile and degraded wood becomes a serious problem worldwide. Therefore, researchers are trying to obtain new materials and capable methods to provide long lasting wood durability. The purpose of such treatment is to improve mechanical resistance of the degraded material, preserving the restored authenticity of the object at the highest level.

Generally, any conservation treatment should follow the internationally established principles: the used treatment for preservation should not alter the object’s integrity and authenticity; the used treatment must have potential reversibility and allow for additional restoration interventions, whenever necessary; the used treatment must remain stable for long periods of time; the used treatment must penetrate and be evenly distributed throughout the wood surface [1,34].

Currently, researchers have proposed tested and analyzed various materials that can be used in wood preservation/restoration, for example, synthetic polymers, such as Paraloid B67, Paraloid B72 and Paraloid B44, and metallic nanoparticles. These synthetic polymers are used in the consolidation process in two phases: the impregnation phase (the polymer solution enters in the wood structure), followed by a conditioning phase (the solid polymer is fixed in the wood structure and the solvents evaporate). The techniques commonly used in a consolidation treatment are total immersion, brushing, spraying, but the technique must be chosen according to the particularities of the object [1,35].

Stone is an inorganic material found in nature and is used for the manufacture of utensils, weapons, jewelry, sculptures and architectural elements. This material represents the oldest traces of human activity, such as in artifacts dating from 700,000 to 130,000 years ago. The first rocks used were granite, diorite, basalt and quartz, due to their high hardness. Later, with the discovery of metals, any type of rock could be processed, for example, marble carvings. Limestone and sandstone are the main materials from which the vast majority of buildings are built; therefore, these types of material are among the most studied for conservation and restoration [3,36].

Over time, different situations have been observed that endanger the integrity of the stone, such as the poor condition of the protection cornices, improper jointing in masonry, temperature fluctuations, rainfall or inefficient precipitation removal [37,38].

The main challenges in the case of stone protection are related to the creation of a hydrophobic surface, its protection against pollutants and the deposition of organic/inorganic particles, while ensuring aesthetic compatibility with the substrate and the reversibility of the treatment [39]. Presently, various materials used for the conservation and restoration of stone have been reported in the literature, such as hydrophobic coatings, antifouling treatments and self-cleaning nanoparticles. Also, nano-HAp was successfully applied to marble and limestone [40], and in a recent study, Ion R.-M. and co-workers successfully applied carbonated hydroxyapatite together with its metal derivatives (silver, strontium, barium, potassium and zinc) on stone models in order to improve the mechanical properties and the resistance to repeated cycles of freeze-thaw [22]. Scherer G.W. tested HAp as a protective treatment for marble against acid rain corrosion, showing an improvement in the marble resistance to the dissolution [41]. Also, Sassoni E. and co-workers investigated the effectiveness and compatibility of HAp treatment for limestone, in comparison with ethyl silicate. It was reported that HAp was able to overcome some ethyl silicate limitations (mainly, prolonging curing time and compatibility), being a very promising consolidant product of porous limestones [42]. The same group evaluated the durability of the HAp treatment to wetting–drying, freezing–thawing and salt weathering cycles, in comparison with ethyl silicates. It was concluded that HAp was a better option for limestone consolidation compared to ethyl silicates, because the samples treated with HAp presented less deterioration than the samples treated with ethyl silicates [43]. 

## 3. Nanomaterials with Applications in the Conservation and Restoration of Cultural Heritage

### 3.1. Main Used Nanomaterials

In the last few years, nanomaterials have been successfully tested to conserve architectural heritage, due to their ability to consolidate and protect damaged building materials. The nanoparticles used for the preservation/restoration of objects have an important role; namely, they cover the surface of the material in order to create a self-cleaning system, preserving the initial appearance of the treated elements, while decreasing the deposition of pollutants and reducing the external degradation processes due to the dirt phenomena. In order to be used in applications of conservation and restoration of the cultural heritage, these nanoparticles must have the following attributes: thermal stability, be biologically and chemically inert, non-toxic, low cost, good adaptability to various environments and good absorption in the solar spectrum [6,11,44,45]. Presently, nanomaterials such as metal nanoparticles (gold, copper and silver) and metal oxides (zinc and aluminum) are widely applied to provide wood protection [46].

In the case of wood, its wall has a porosity of dimensions on a molecular scale due to the partial filling of the spaces between the cellulose microfibrils. Small nanoparticles can deeply and efficiently penetrate into wood, changing the chemistry of the wood surface and improving its properties. In addition, full penetration and even distribution are obtained if the nanoparticle size is smaller than the pore diameter in the wood wall [1,46]. It has been reported that a thin and homogeneous layer of TiO_2_ nanoparticles, with an average size of 50 nm, covers the internal structures of the wood, without changing its natural appearance [11]. In Table 1, the main nanomaterials used for wood consolidation and their properties are presented.

### 3.2. Physico-Chemical and Mechanical Properties

In the last decade, nanotechnologies have become a key factor in the field of cultural heritage, due to their ideal properties that help to protect heritage objects, for example: cleaning surfaces, acting against microorganisms, protecting materials from the negative effects of UV radiation, etc. The advantage of applying nanomaterials is represented by the possibility of obtaining a great depth of penetration in the structure of the materials (mainly it varies according to the porosity and the moisture content of the material) and a high efficiency, preserving the original material [6,44,57].

Titanium dioxide (TiO_2_) is an inorganic material that is found in the form of nanocrystals or nanogranules, and is intensely used in various applications due to its properties. In recent decades, this material has been intensely used as a pigment, in UV protection, paints, ointments, toothpaste, etc. This material has many advantages that make it ideal for different applications, such as the high surface area offered by the small size of the TiO_2_ particles or the increased antimicrobial activity. These two characteristics are closely related to the crystalline structure, shape and size of nanomaterials [58,59].

TiO_2_-based nanomaterials are obtained by various methods, such as sol-gel, the hydrothermal method, the direct oxidation method, chemical vapor deposition, electrodeposition, etc. [58]. Depending on the method of preparation, TiO_2_ can be obtained in various forms, including nanoparticles, nanofibers and nanotubes [58,60]. The most widely used process for obtaining these nanomaterials is the sol-gel method. In a typical sol-gel process, a colloidal suspension (sol) is formed from the hydrolysis and polymerization reactions of the precursors, and then the complete polymerization and loss of the solvent leading to the passage from the liquid sol to a solid gel phase occurs. TiO_2_ nanomaterials are synthesized by the sol-gel method of hydrolysis of a titanium precursor [61,62,63,64,65]. Sugimoto T. and co-workers obtained different shapes and sizes of TiO_2_ nanoparticles by varying the reaction parameters. Thus, it was observed that the morphology of the TiO_2_ nanoparticles changes from cubic to ellipsoidal, when the solution is brought to a pH above 11 with triethanolamine (it acts as a surfactant). When diethylenetriamine was used, at pH above 9.5, it was observed that the shape of TiO_2_ nanoparticles evolves in ellipsoids with a higher appearance than that with triethanolamine [66]. Also, it has been reported in the literature that the shape of TiO_2_ nanoparticles can be changed from a round to a cubic form by using sodium oleate and sodium stearate. The shape control is attributed to the growth rate regulation of the different crystalline planes of the TiO_2_ nanoparticles by the specific adsorption of the shape regulators to these planes under different pH conditions [66,67,68]. In another study, TiO_2_ nanoparticles were obtained by the sol-gel method, and subsequently, the influence of acid pH (3.2–6.8 with a hydrochloric acid solution) on the formation of TiO_2_ nanocrystalline powders and photoluminescent properties was studied. Thus, it was reported that the best luminescence property was obtained for the TiO_2_ nanoparticles synthesized at a pH of 5.0 [69].

Presently, TiO_2_-based nanomaterials have been tested on different materials in order to consolidate or restore the cultural heritage. For example, La Russa M.F. and co-workers studied the efficiency of TiO_2_ nanoparticles dispersed in an acrylic polymer solution on limestone and marble samples. The biocidal efficacy of TiO_2_ against *A. Niger* fungi was evaluated, and the results show a high efficiency of growth inhibition on both types of samples. Also, the photodegradation tests revealed the efficiency of increasing the speed of oxidation of the methylene blue stains. After aging, the behavior of the two types of samples was different. The treated limestone surfaces do not appear to be affected by solar radiation, while in the case of marble, the coating is almost inefficient after aging, suggesting that the composition of the sample also plays an important role [70]. In another study, De Filpo G. and co-workers tested these nanomaterials in order to study their efficiency as antifungal and biocidal agents for woodworking. After the wood samples were treated, they were put in contact with two species of fungi, *Hypocrea lixii* and *Mucor circinelloides*, which are known to be responsible for the rapid degradation of the wood. The results show that the photo-catalytic activity of TiO_2_ nanoparticles prevents fungal colonization of wood samples for a much longer period, compared to untreated samples [71]. Also, this treatment was deposited on the wood surfaces using a plasma jet with atmospheric pressure in order to improve the stability of the wood against the ultraviolet (UV) light and humidity resistance capacities. Color changes during UV exposure for both uncoated and coated wood samples were measured. It was observed that the sample coated with TiO_2_ had become more resistant to color change after exposure to UV radiation than untreated wood [72].

Zinc oxide (ZnO) is an inorganic compound used in various applications: pharmaceutical industry, cosmetics, chemicals, ceramics, paint and glass [73]. The antibacterial activity of ZnO increases with decreasing particle size and such action can be stimulated by visible light. Also, the absorption property of UV radiation improves the stability of the composite [74].

Currently, various obtained methods involving ZnO-based nanomaterials have been developed, such as: vapor phase growth, the vapor-liquid-solid process, electrophoretic deposition, sol-gel processes, homogeneous precipitation, etc. The properties of ZnO nanoparticles, such as crystallinity and morphology, can be controlled by adjusting factors such as pH, reaction temperature, time and solvent [75].

Like TiO_2_, zinc oxide has been intensively studied as a potential enhancer used in conservation and restoration applications of cultural heritage, thus demonstrating that it can be used successfully in this field. For example, the humidity of the ZnO surfaces was examined; these flat ZnO substrates had a water contact angle of up to 109°, confirming the material’s ability to provide hydrophobicity to the surface on which it is applied [76]. In another study, pine samples were treated with ZnO nanoparticles in order to investigate the efficiency of the consolidant for water absorption. It has been reported that ZnO nanoparticles (at a concentration greater than or equal to 2.5%) significantly improved water absorption resistance for 12 months of outdoor exposure, compared to control [50]. Also, David M.E. and co-workers confirmed the capacity of cellulose acetate-based micronized particles (ZnO and TiO_2_) to provide hydrophobicity on the pinewood surface [77]. Clausen C.A. and the co-workers reported that the use of ZnO nanoparticles as a wood coating product leads to a significant decrease (under 4%) in the consumption of eastern subterranean termites in the case of wood blocks and to an increase in the mortality of these termites by over 94% [78]. In another study, it was reported that delamination between the wood surface and the coating layer caused by UV and moisture irradiation could be avoided using ZnO nanoparticles [79].

Gold (Au) nanoparticles are intensively used in various applications due to their unique properties and increased surface functionalization ability with different compounds [80]. These nanoparticles can be synthesized into different shapes and sizes due to the various synthesis methods available. The most commonly used methods for the synthesis of Au nanoparticles are chemical and biological methods, but the chemical method offers the advantage of better control over the size and shape of the nanoparticles. Since the size and shape of the nanoparticles depend on the synthesis method and it can be controlled by adjusting the reaction parameters (including temperature, concentration and pH) [81,82]. The physical, chemical and optical properties are strongly influenced by the size of the nanoparticles. For example, it has been shown that Au nanoparticles with dimensions below 100 nm provide a large surface-volume ratio, and the chemical, physical and optical properties are different from those of the same bulk material [83,84]. Also, another important advantage of these nanoparticles are the microbiological properties, these nanoparticles presenting an increased antimicrobial, antifungal and antibiofilm activity [85,86].

Due to these properties, Au nanoparticles have attracted the attention to be studied in different fields, including the conservation and restoration of cultural heritage. For example, Ion R.M. and co-workers studied the efficacy of a new system based on Au and hydroxyapatite (HAp) nanoparticles on hazelnut wood samples. Following the obtained results, it was observed that the wood samples treated with the Au and HAp system were more stable and have superior mechanical and hydrophobic properties, compared to the samples treated with either Au, HAp or untreated (control) samples. These improved properties are mainly due to the presence of gold nanoparticles, which have the ability to insert/penetrate into the wood channels, thus leading to hardening and protecting the wood (Figure 3). Also, compared to HAp, the surface hardness increased considerably in the case of the sample on which the Au and HAp based system was applied, due to the network of fibers generated on the wood surface [1]. 

Silver (Ag) nanoparticles and their nanocomposites are among the most used nanomaterials, being used in almost all types of applications (medicine, antimicrobial agents, sensors, catalysts, coating agents, cosmetics, water treatment, etc.) [87,88,89]. Of the entire range of metallic nanoparticles, Ag nanoparticles are the most widely used due to their unique physical, chemical and biological properties. The advantage of these nanoparticles as compared to the other noble metals, in terms of their physico-chemical properties are: high electrical and thermal conductivity, non-toxicity, stability under environmental conditions, low costs of obtaining, wide absorption of light, chemical stability and catalytic activity. Moreover, they have a broad spectrum of high antimicrobial activity (bactericidal and fungicidal activity) [90,91]. Currently, Ag nanoparticles can be synthesized by physical, chemical and biological methods. Physical and chemical methods are the most laborious, compared to biological synthesis, which consists of a fairly simple and easy process to produce large-scale nanoparticles that have attractive properties, such as high yield, solubility and stability [89,92].

As in the case of Au nanoparticles, the properties of Ag nanoparticles depend on the size and shape of the nanoparticles, characteristics that are closely dependent on the synthesis method (they can be controlled by adjusting the reaction parameters). For example, it has been reported that the bactericidal activity of Ag nanoparticles is strongly influenced by nanoparticle size; the smaller the nanoparticle size, the stronger the bactericidal activity is [93,94].

In recent years, Ag nanoparticles have been intensively studied to demonstrate their efficiency as a consolidant for the conservation and restoration of cultural heritage. In a study, Berrocal A. and co-workers tested a solution of Ag nanoparticles (50 ppm (parts per million)), which was incorporated by pressure into three commercial wood species (*Acacia mangium*, *Cedrela odorata* and *Vochysia guatemalensis*) from Costa Rica. Following the experiment, it was observed that the mass loss was less than 5% for all three types of wood treated with Ag nanoparticles, while untreated wood showed a mass loss of more than 20%. Also, the water absorption capacity was reduced for the three wood species treated with Ag nanoparticles, while the dimensional stability increased for two of the three wood types (*Cedrela odorata* and *Vochysia guatemalensis*) [95]. 

In another study, Mantanis G. and Papadopoulos A.N. studied the improvement of pine wood from the point of view of water absorption using Ag nanoparticles. It has been observed that the immersion of pine wood in an Ag nanoparticle solution for 2 min significantly reduces water absorption, thus confirming the ability of this material to be used as a consolidate in cultural heritage [96].

Magnesium oxide (MgO) is a basic oxide and can be formed by the reaction of metal with oxygen gas [97]. MgO nanoparticles can be synthesized by various methods, such as sol-gel, thermal decomposition, precipitation reactions, etc. [45]. Currently, magnesium oxide has various applications, such as an adsorbent for chemical agents, photocatalyst, antibacterial agent, antioxidant agent, etc. Comparing the antimicrobial activity of MgO nanoparticles on Gram-positive and Gram-negative bacteria, it has been shown that this material is more effective against Gram-positive bacteria (*S. aureus*). It has also been shown that these nanoparticles possess photocatalytic activity both under UV irradiation and in sunlight, which opens up several possible areas of application of these ecological MgO nanoparticles [98,99,100].

In the last few years, MgO nanoparticles have been tested as potential consolidates for various materials (wood, paper, etc.). In one study, it was reported that MgO is an excellent paper de-acidification agent, which ensures good physico-chemical compatibility with the substrate, without producing undesirable side effects on the treated material [45]. Also, Castillo I.F. and co-workers treated old paper with MgO nanoparticles to prevent fungal damage of paper artifacts from fungi commonly found on the surface of old colonizing paper: *A. niger*, *C. cladosporioides* and *T. reesei*. After treatment, it was observed that dispersions of MgO nanoparticles on original paper samples from the 18th century were effective in preventing fungal colonization without altering the appearance of paper artifacts. In addition, this treatment inhibited activity in *A. niger* and *T. reesei* fungi [101].

Iron oxides are found in a wide variety of structures and are present in many types of applications, from geological to nano-technological applications. They have seven crystalline phases, the most common being α-Fe_2_O_3_ (hematite), γ-Fe_2_O_3_ (maghemite) and Fe_3_O_4_ (magnetite). Due to their ideal properties (magnetic, electrical, optical), all these oxides have been extensively investigated by chemists, engineers and physicists. Iron oxide nanomaterials can be synthesized by different methods (co-precipitation, hydrothermal method, microemulsion, thermal decomposition, sol-gel, etc.) in different shapes and sizes, such as nanotubes, nanowires, fibers and rings. The performance of iron oxides is strictly influenced by their morphology, size and porosity [102,103,104]. These nanomaterials have been successfully used in many applications, such as medicine, water cleaning, cathodes, photoelectrochemical systems, dyes, etc. [103,105]. These nanomaterials have also been used to protect and improve the resistance of wood. Schauwecker C.F. and the co-workers used iron oxide particles with different crystalline shapes and sizes to protect against the discoloration of wood under solar radiation. The results of this study suggest that the large size of the oxide particles offered both greater opacity and protection [106]. Another advantage of these transparent iron oxides is that they are multifunctional non-toxic pigments, which can be combined with a range of color shades with excellent UV absorption, transparency and weather stability; factors that lead to increased protection of the wood [107]. 

Silicon dioxide (SiO_2_) is a colorless, crystalline solid substance that does not react with water and is resistant to acids. SiO_2_ particles can be used as a source of material for photovoltaic cells, semiconductor electronic devices, catalysts, film substrates, ceramics, plastics fillers, humidity sensors, absorbents, anti-corrosive agents, etc.

The synthesis of SiO_2_ particles can be realized by several methods, such as chemical vapor deposition, plasma and combustion synthesis, the hydrothermal method and sol-gel processing. Among these synthesis methods, sol-gel has low cost advantages in processing and also facilitates the control of SiO_2_ properties (purity, composition and material homogeneity) [108,109,110]. The SiO_2_ properties are dependent on the conditions of its synthesis, such as temperature, pH, washing and drying modes. These factors influence the size of the SiO_2_ particles, their aggregation and the specific surface area [109].

SiO_2_ nanoparticles were also used in coatings for materials belonging to the cultural heritage. Doubek S. V. and co-workers investigated the SiO_2_ capacity on beech (*Fagus sylvatica*) and fir (*Abies alba*) wood. Following the treatment, it has been observed that silica mineralization can improve some of the technical properties of the wood, leading to the prolongation of the life of the wooden structures. It has also been reported that silicon dioxide has not shown anti-mold efficiency, a property that is essential in protecting wood [111]. In another study, it was observed that the SiO_2_ nanoparticles treatment led to a significant reduction in the swelling of the wood, but this also decreased the resistance of the wood [112]. 

The effect of the properties of SiO_2_ nanoparticles on the wood surface was also studied by Fu Y. and co-workers. The results show that the water absorption and the hygroscopic expansion rate of the treated wood were lower, compared to the control sample. Also, the resistant properties to discoloration were improved by 1.5 times, compared to the control sample. The contact angle test showed that the treated wood was more hydrophobic than that of the control sample and increased with the time of immersion in the SiO_2_ solution. Also, the treated wood showed increased resistance to aging compared to the control sample [113]. These studies confirmed that SiO_2_ can be considered a potential material to be used as a consolidant in wood preservation and restoration, but it is necessary to improve the properties of this material to obtain better performances (e.g., SiO_2_/Ag, SiO_2_/TiO_2_ for anti-mold properties [71,114]). 

Copper oxide (CuO) is an inorganic compound produced on a large scale due to the various applications in which it is used successfully. Copper oxide is a semiconductor metal with unique optical, electrical and magnetic properties and has been used for various applications, such as the development of supercapacitors, infrared filters, magnetic storage media, sensors, catalysts, semiconductors, etc. [115,116]. One of the most important parameters in the synthesis of CuO nanoparticles is the control of the particle size, their morphology and crystallinity, and different methods of synthesis have been developed to achieve this objective; several more investigated approaches include the sonochemical method, the sol-gel method, laser ablation, the electrochemical method and surfactant techniques [115,117,118].

The synthesis methods of CuO nanoparticles have advanced significantly in the last ten years due to the various important industrial applications. The synthesis technique is important for the properties of the final nanosystem, because it can control the size and morphology of the nanoparticles. Also, these nanoparticles exhibit various optical, magnetic, mechanical and electrical resistivity properties, which differ from the characteristics of the bulk solid material [115].

In the last few years, metal nanoparticles have been intensively studied as wood protection consolidates. Shiny K. and co-workers obtained CuO nanoparticles using plant extracts (Neem leaves (*Azadirachta indica*), Pongamia (*Pongamia pinnata*), Lantana (*Lantana camara*) and orange peel extract (*Citrus reticulata*)), which are known to have wood preservation properties. The effectiveness of the obtained CuO nanoparticles was tested against wood termites. It has been found that all CuO nanoparticles obtained from plant extract provide termite protection for a period of six months, compared with control. The development of a synthesis pathway based on sustainable plant extracts for metallic nanoparticles allows the possibility of combining the intrinsic property of the plant extract and the metallic nanoparticles for a potential application in wood protection. The resulting formulation can protect the wood from biodeterioration in a more efficient way, without damaging the environment [119]. In another study, Akhtari M. and D. Nicholas tested and compared the effectiveness of CuO and ZnO nanoparticles as potential wood consolidants. CuO nanoparticles have been found to be much more effective in protecting wood against termites, compared to ZnO nanoparticles [120]. 

CuO nanoparticles have been used to protect wood against various microorganisms (fungi and bacteria) [121,122]. Also, another important advantage of using the CuO nanoparticles as a wood consolidant is the fact that the effective removal of the CuO layer from the wood surface was achieved with the help of chelated agents, without causing damage to the treated wood initially [123].

## 4. Tubular Nanomaterials with Applications in the Conservation and Restoration of Cultural Heritage

In the last few years, tubular nanomaterials (like carbon nanotubes, titanium oxide nanotubes, ZnO nanotubes, etc.) have attracted attention for use in different fields due to their structures, as well as their ability to present multiple walls. In addition, their unique physical and chemical properties, their interior voids and exterior surfaces make them ideal candidates for various applications [124,125,126,127,128,129,130].

### 4.1. Structure and Synthesis Methods of CNTs

Carbon nanotubes (CNTs) were discovered in 1991 by Sumio Iijima, through the action of a catalyst on the gaseous species created by the thermal decomposition of hydrocarbons. Subsequently, multi-walled carbon nanotubes (MWCNTs) were obtained as by-products for the production of fullerenes in an electric arc without catalysts. Two years later, after repeated attempts to “fill” CNTs with various metals, single-wall nanotubes (SWCNTs) were discovered.

CNTs are large cylindrical molecules consisting of a hexagonal arrangement of hybridized carbon atoms, which can be formed by rolling a single graphene sheet (SWCNTs—Figure 4A) or rolling two or more sheets of graphene linked through van der Waals non-covalent force, that acts between the carbon atoms of the different walls (MWCNTs—Figure 4B). Typically, these nanotubes are coated at both ends with a hemispherical arrangement of carbon networks, called fullerene [131,132,133,134,135].

These two types of CNT differ not only in terms of structure, but also due to their properties; for example, SWCNTs have important electrical properties (they can be excellent conductors), compared to MWCNTs. SWCNTs are still very expensive, and the development of more cost-effective synthesis techniques is vital for their use in a wider range of applications [136]. 

CNTs are widely used in many applications due to their unique properties (electrical, mechanical, optical, thermal and other properties). The application of CNTs is usually given by their structure (number of walls, diameter, length, chiral angle, purity, structural quality, etc.), which gives the specific properties. These carbon nanotubes can be synthesized by various methods and each of these methods has some advantages and disadvantages that lead to obtaining CNTs with different properties [137,138].

For example, a sonochemical method was also used to synthesize SWCNTs by the ultrasonic irradiation of a solution containing silica, ferrocene and p-xylene powders. In this synthesis, ferrocene was used as a precursor for the Fe catalyst, p-xylene was used as a carbon precursor and the silica powder provided nuclear sites for the growth of CNTs. Ferrocene was sonochemically decomposed to form small Fe groups, and p-xylene was pyrolysed to carbon atoms and carbon moieties. This approach provides a convenient synthetic route for preparing CNTs under environmental conditions. In addition, no extra purification steps were required in this process, which opens up the possibility of large-scale ultrasonic synthesis of SWCNTs [139]. In another study, Wang W. and co-workers synthesized MWCNTs with outer diameters between 9 and 19 nm and inner diameters between 4 and 8 nm by decomposing polyethylene glycol (PEG) into a basic aqueous solution with high NaOH concentration under hydrothermal conditions at a temperature of 160 °C, without the addition of a Fe/Co/Ni catalyst [140]. In another study, Jagadish K. and co-workers synthesized MWCNTs by the hydrothermal method of polystyrene (PS) in the presence of the catalyst (Fe particles) at a temperature of 400 °C. Following the morphological characterization, it was observed that the obtained MWCNTs had inner and outer diameters of 19 and 22 nm, respectively, with a wall thickness of 5 nm, their length being a few millimeters [141]. Also, Manafi S. and co-workers prepared MWCNTs by an easy sonochemical/hydrothermal method. MWCNTs were fabricated hydrothermally using dichloromethane, cobalt chloride and lithium as starting materials in an aqueous NaOH solution, and the ultrasonic pre-treatment of the solution mixture was performed before hydrothermal conditions (150–180 °C for 24 h). Following this synthesis, high purity MWCNTs with lengths of 2–5 μm and diameters of 60 ± 20 nm were obtained [142]. 

In another study, Chrzanowska J. and co-workers obtained SWCNTs by the laser ablation method (laser source Nd: YAG–neodymium-doped yttrium aluminum garnet), and observed that the properties of the synthesized CNTs are highly dependent on the laser fluency. In the case of the laser with a wavelength of 355 nm, the best SWCNTs (morphologically) were obtained at the fluence of F = 3 J* cm^−2^, while for 1064 nm, good results were obtained in the fluorescence range of 1 ≤ F ≤ J* cm^−2^ (the distribution of nanotubes being the smallest at this fluency, with a diameter of 1.3 nm). Also, it has been observed that the distribution of nanotubes, in terms of diameters, became wider as the fluency increases [143]. 

Wu H. and co-workers tried to obtain CNTs by the electrochemical reduction of CO_2_ in different molten Li-Na-K carbonate mixtures. By adjusting the density of the electric current, the electrolyte and the temperature, the carbon products had different morphologies of structures. It was observed that in the case of the Li-Na-K electrolyte, no CNTs were formed, but a high efficiency was observed for pure Li, Li-Na or small Li-Ba and carbon electrolytes. Also, it was observed that the diameter of the CNTs increased with the increase in the electrolysis time [144]. In another study, Dimitrov A.T. and co-workers reported the obtaining of MWCNTs with lengths between 50 nm and 3 μm and diameters between 10 and 80 nm, by electrolysis in Li−Cl molten mixtures [145].

In another study, MWCNTs of about 16–55 nm in diameter and 1.2 µm in length were obtained by chemical vapor deposition (CVD) method using ferrocene and molybdenum hexacarbonyl as precursors of catalyst nanoparticles and methane as a nontoxic and economical carbon source [146]. Seidel R. and co-workers also tried to obtain SWCNTs at the lowest possible temperature. Thus, dense SWCNT networks were synthesized by the thermal CVD method, at temperatures up to 600 °C using Ni catalyst layers with a thickness of about 0.2 nm [147]. 

Chemical synthesis is a new technique for obtaining CNTs, being adapted according to the Staudenmaier method [148] and one of the most promising routes for the production of MWCNTs. This technique allows the easy and cheap preparation of MWCNTs, at low temperatures (70 °C) and without applying pressure. The obtaining of CNTs by this technique is incipient, with only a few articles being reported in the literature.

Using this method, Lee D.W. and Seo J.W. obtained MWCNTs with a diameter of about 14 nm and with a zig-zag structure [149]. In another study, Ali B. and co-workers synthesized MWCNTs by chemical synthesis with a diameter of about 13 nm and high thermal stability [150].

### 4.2. Properties of CNTs Important for Cultural Heritage

Over the years, CNTs have been used in many fields of application due to their highly useful physical, chemical and mechanical properties. For example, CNTs introduced into polymeric matrices, such as epoxy, are a new generation of composite materials with advanced mechanical properties [151]. It has been reported that the introduction of SWCNTs, and MWCNTs into polymeric matrices showed significant increases in resistance and Young’s modulus compared to control (polymeric matrices) [152]. Therefore, CNTs are suitable candidates to be used in conservation and restoration of cultural heritage due to their ideal properties:

Mechanical and elastic properties: CNTs are stiffer than steel and are highly resistant to the application of physical forces (pressing on the top of a nanotube will cause it to bend, without damaging the top) [153,154]. Compared to graphite fibers, CNTs exhibit tensile strength and superior elastic behavior. Graphite fibers have been reported to have a tensile strength of approximately 1 GPa, while MWCNTs have an average of 14.2 ± 0.8 GPa [155], and SWCNTs average 3.66 ± 0.4 GPa [156]. Analyzing the elasticity modulus (or Young’s Modulus), it was reported that SWCNTs have a value of about 1 TPa, and MWCNTs have a higher modulus of elasticity of about 1.28 TPa [152]. These properties differ depending on the obtaining method of the nanotubes and the temperature at which the CNTs are synthesized, respectively. Therefore, the extreme mechanical strength of CNTs makes them the best known material, with great potential for applications that require high mechanical strength materials. The use of CNTs in order to improve the mechanical properties of different materials has been reported in several studies [157,158,159].

Optical properties are very important because they show how the material interacts with light. Presently, there are few studies in the literature on the optical properties of CNTs. In a recent study, it was reported that UV exposure of these CNTs has no direct impact on their properties [160]. For this reason, CNTs have been introduced into the polymeric matrix (poly (methyl methacrylate)), noting that the addition of nanotubes into the matrix was beneficial for improving its resistance against UV radiation. It was concluded that the high-energy radiation was dissipated through the nanocomposite CNT network, thus improving the tolerance to deterioration [161]. Nguyen T. and co-workers investigated the release capacity of CNTs from the polymer matrix in the environment after UV exposure. Thus, after obtaining an epoxy nanocomposite based on MWCNTs, it was irradiated with UV light (with a maximum value of 4865 MJ/m^2^) at different doses, and the effects of UV exposure on surface accumulation and the potential release of MWCNTs has been studied. After UV exposure, it was observed that the nanocomposite matrix suffered photodegradation, leading to the formation of a dense network structure of MWCNTs, but no release of nanotubes was detected in the environment, even at very high UV doses [162]. Also, Petersen E. J. and co-workers reported that the presence of CNTs on the sample surface reduces the affinity of the epoxy matrix to photodegradation. Before the UV exposure, a smooth and flat surface of the nanocomposite can be observed, and after the UV exposure (1089 MJ/m^2^), it can be observed that the surface morphology was significantly modified, having a hard, non-uniform appearance, which attests that the upper layer of the polymer degraded, being left on the surface the nanotubes [163]. These studies confirmed the capacity of the CNTs to protect objects against photodegradation.

Hydrophobic properties: this property is indispensable in cultural heritage protection and CNTs’ hydrophobicity has been reported in several studies. For example, Eseev M. and co-workers obtained superhydrophobic surfaces by applying the MWCNT agglomerates made from xerogel milled by the carbon band CVD method and by studying the functionalization time of the MWCNTs. Thus, they obtained a material with a contact angle of almost 151° and concluded that the short functionalization periods of MWCNTs (up to 90 min) are dominated by the cleaning process of the nanotubes (amorphous carbon and metal nanoparticles—the catalyst). The MWCNTs functionalized for 90 min showed an increase in the contact angle from 112° (value obtained for a working time below 90 min) to 140° (90 min operating time), allowing this coating to be considered as being highly hydrophobic [164]. In a recent study, MWCNTs were synthesized by catalytic CVD, and on the surface of these nanotubes was physically adsorbed fluid poly (dimethylsiloxane) fluid, in order to obtain a superhydrophobic nanocomposite material. After analyzing the contact angle, a value of 152° was reported for the nanocomposite, while for the poly (dimethylsiloxane), a value of 57° was obtained. These results confirm once again the high hydrophobicity of CNTs [165]. The presence of these nanotubes in the polymer matrix can lead to significant decreases in the water adsorption capacity of the polymeric matrix, due to the non-polar hydrophobic structure of the CNTs [166].

### 4.3. CNTs Used in Conservation and Restoration of Cultural Heritage and Future Challenges in This Area

Potential applications of CNTs include obtaining new materials, optical and microelectronic devices, textiles and clothing. Presently, the aim is to create hydrophobic, anti-freeze and self-cleaning coatings in several industrial sectors, particularly in the cultural heritage area. These coatings will also have a wide application in the construction and automotive industries [164]. To date, only a few studies have been reported on heritage conservation by using CNTs, which have been very successful in the consolidation and restoration of the object [167,168]. For example, Cestari B. C. and co-workers reported the successful use of CNTs embedded in epoxy resin as coatings for timber structures. It has been proven that the use of these materials as timber consolidate leads to a significantly improved in mechanical strength of the wood [169]. Valentini F. and co-workers sustained the use of CNTs in cultural heritage by reporting the efficiency of SWCNTs in the removal of the black crust from the surfaces of pentelic marbles from the Basilica Neptuni (Rome, Italy) [170]. The capacity of CNTs in the consolidation and restoration of stone materials was also tested by mixing MWCNTs with mortar. It was observed that the addition of MWCNTs in mortar improves its mechanical properties. Additionally, the results show that the MWCNT concentration plays an important role in the matrix reinforcement capability; around 0.2% wt. of MWCNTs are needed to better reinforce the matrix in terms of increased flexural strength and to achieve optimum properties in terms of piezoresistivity [171,172,173]. In another study, CNTs were used in an innovative and precise instrument for mild heating, designed specifically for art conservation, in the form of lightweight, flexible, transparent and breathable film-like mats. This device is made by various layers, and the main layers consist of a substrate (transparent polyester film-gives good adhesion to any kind of coating), the conductive CNT coating, a protective-insulating coating on top and two copper electrodes glued on polyester film (for the electrical supply). The device is driven by a programmable mobile touch screen console, which gives the operator ultimate control over the temperature and heating pattern, which is unprecedented in art conservation. In this study, the device was successfully applied on paintings, textiles and works on paper and the humidification of the artifacts was maintained in the required parameters, leading to a better preservation of objects over time. Apart from the advantages mentioned above, this device offers the possibility to be designed in ultra-thin, transparent, and woven forms, and also as gas permeable membranes to permit the migration of vapors. This treatment can therefore deal with local, global or specific conservation problems, allowing conservators to easily apply mild heating locally and over very large areas [25]. In a recent study, Eseev M. and co-workers obtained superhydrophobic anti-icing coatings, prepared from milled xerogel based on CNTs and then, the resulting materials were applied on the steel sample surfaces. It was reported that the treated sample presented a water contact angle of approximately 156° at the droplet inclination angle of 3.6°, confirming the superhydrophobicity of the coating. Moreover, the high hydrophobicity of the coating allows the deceleration of frosting formation and the droplet slips from the coating before it can freeze. The authors also suggested that, due to the high sorption capability of the CNTs, lubricants can be loaded on the CNTs surface. The suggested approach can be used to create superhydrophobic coatings, which can be used in the cultural heritage field, for anti-freezing protection, but also for protecting other surfaces, such as aircraft surfaces [174]. Also, Krishnamurthy M. and co-workers reported the superhydrophobicity of the coating based on the polymerization of SWCNTs with thiophene/aryl compounds tested on leather and glass surfaces, with a water contact angle of 155° and 163°, respectively [175].

Due to the promising results obtained for CNTs incorporated in polymeric matrices, in recent years, a new concept has been tried, namely the functionalization of CNTs with metal nanoparticles. For the functionalization of CNTs, several methods have been developed, including covalent modifications (based on the formation of a covalent bond between functional entities and the CNT framework), non-covalent modifications (using various adsorption forces, such as Van der Waals forces, hydrogen bonds, etc.) or electrostatic interactions. Covalent bonding between nanoparticles and CNTs was obtained by various acid treatments to create bonding groups such as carboxyl (–COOH), carbonyl (–C=O) and hydroxyl (–OH); however, the mechanical and electronic properties of CNTs can be significantly degraded after acid treatment due to the introduction of defects. The non-covalent attachment of nanoparticles is more able to preserve the unique properties of CNTs with reduced defects [176]. Depending on the methods used, the functional groups/materials can be introduced on the surface of CNTs. Functionalized CNTs can have mechanical, optical or electrical properties that are different from those of the initial nanotube and significantly improved [177,178]. In a recent study, the functionalization of MWCNTs with Au nanoparticles in three stages was successfully performed: functionalization of the surface of carbon nanotubes; grafting of cysteamine hydrochloride by a thiol reaction and decorating the gold nanoparticles by chemical covalent bonds. After functionalization, it can be observed that Au nanoparticles attached to the surface of MWCNTs have spheroidal shapes, with dimensions between 15 and 35 nm [179]. Also, Ag nanoparticles have been successfully linked on the surface of CNTs, thus expanding the applicability of CNTs in various fields, such as medicine, food industry, controlled release systems, anti-cancer drugs, antimicrobial agents, coating for different surfaces and textile industry [180]. Table 2 presents the structures based on functionalized CNTs and the areas in which these structures were used.

Presently, the application of these functionalized CNTs in the cultural heritage area was not achieved, but studies from other areas confirmed that functionalized CNTs offer superior mechanical, optical or electrical properties, compared with non-functionalized CNTs. Therefore, the presence of CNTs in the field of heritage can bring important improvements in the consolidators, these nanotubes having the most important properties necessary for this area.

## 5. Conclusions and Future Perspectives

In the last few decades, the conservation of cultural heritage has become a topic of interest worldwide, due to the need to preserve the authenticity of artifacts and constructions, as well as the history of mankind. Classic examples of artifacts include stone tools, wooden tools and objects, metal or personal ornaments and ceramic vessels. Due to the age of these objects and the external degradation factors, their structure is severely affected. For example, wood materials are constantly subject to several serious degradation factors, such as biological or chemical degradation, which more or less affect the structural integrity and mechanical strength of these materials.

Currently, nanomaterials are a good solution for this problem of cultural heritage, due to their ability to consolidate and protect damaged building materials. Nanomaterials used in cultural heritage conservation and restoration applications must meet the following requirements: to have thermal stability, to be biologically and chemically inert, to be non-toxic, to have a low cost, to have good adaptability to various media and good absorption in the solar spectrum. Also, once applied to the surface of objects for preservation and restoration, these nanomaterials must create a self-cleaning system, thus preserving the initial appearance of the treated elements, while decreasing the deposition of pollutants and reducing the onset of external degradation processes due to contamination. Currently, various nanomaterials are successfully used to preserve and restore heritage objects, such as metal nanoparticles (gold, copper and silver) and metal oxides (zinc, titanium, iron and aluminum).

In recent years, tubular nanomaterials have attracted attention for use in different fields due to their structures, as well as their ability to present multiple walls. In addition, compared to their unique physical and chemical properties, their interior voids and exterior surfaces make them ideal candidates for various applications, including for conservation and restoration applications of cultural heritage. The carbon nanotubes are among examples of nanotubular materials, which have the advantage on being synthesized by using several methods (e.g., arc discharge, sonochemical or hydrothermal method, laser ablation, etc.). These nanotubes have properties necessary for the preservation and restoration of objects, such as superior mechanical and elastic strength, high hydrophobicity, optical properties, large specific surface area for absorption of other nanomaterials, relatively good biocompatibility, etc.

Presently, both the introduction into a polymeric matrix of CNTs and the attachment of metal nanoparticles on the surface of CNTs has been successfully achieved. These obtained composites were successfully applied in various applications, but also for the conservation and restoration of objects. It was observed that following the introduction of CNTs in a polymeric matrix, the preservation and restoration capacity of the object treated with this consolidant has increased significantly, with the treated object presenting superior mechanical properties and a better UV resistance. Also, after CNT functionalization and their application as coating for objects, mechanical, optical or electrical properties were achieved that were significantly improved than those of the original nanotube (without functionalization). Therefore, the presence of CNTs in the field of heritage can bring important improvements in the consolidators, with these nanotubes having the most important properties necessary for this area.

## Figures and Tables

**Figure 1 materials-13-02064-f001:**
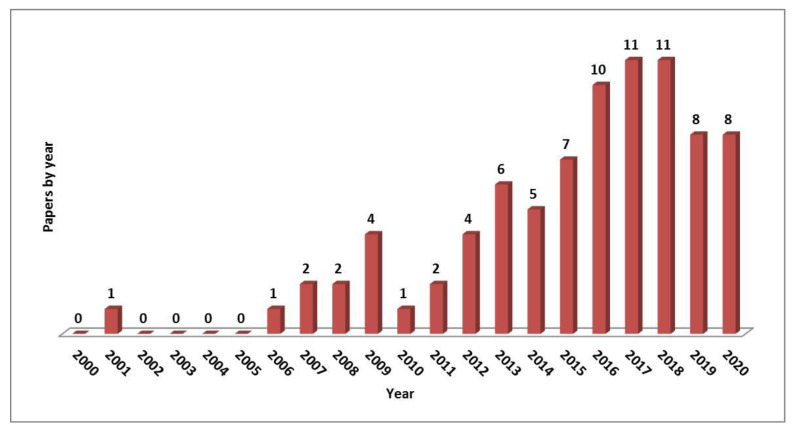
Graph of scientific papers published per year in the period 2000–2020 regarding nanoparticles used to consolidate different artifacts. From Scopus (https://www.scopus.com).

**Figure 2 materials-13-02064-f002:**
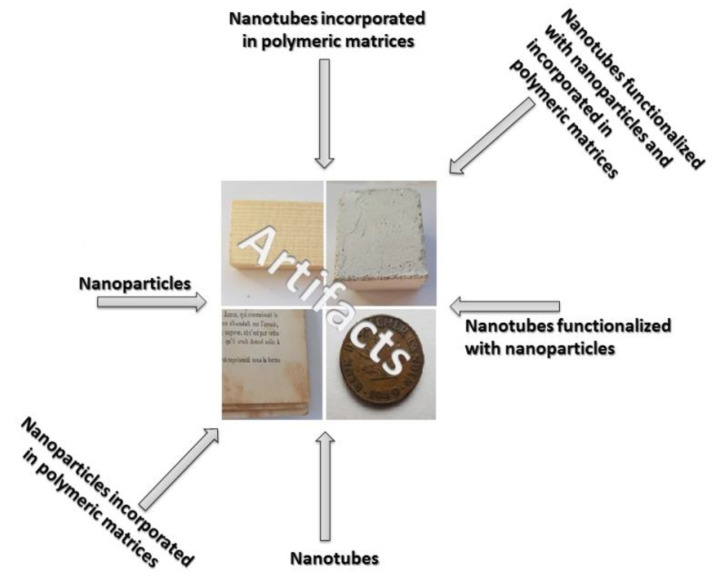
Nanomaterials used as consolidates in conservation and restoration of cultural heritage.

**Figure 3 materials-13-02064-f003:**
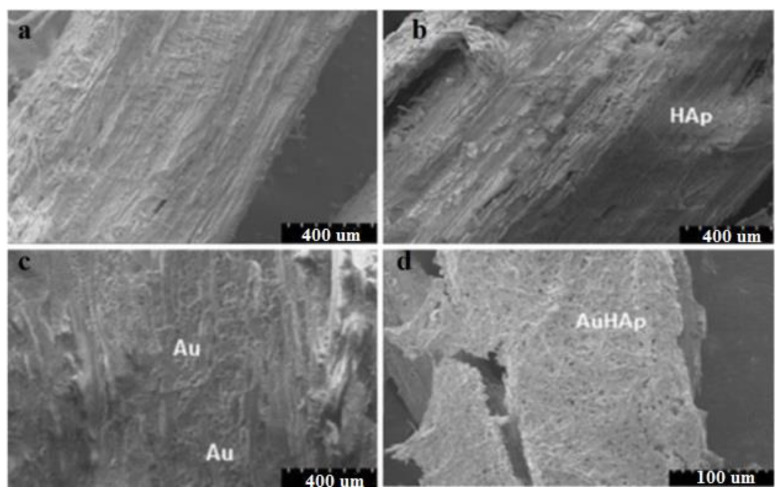
SEM images of the treated wood: (**a**) old wood, (**b**) old wood treated with HAp, (**c**) old wood treated with Au, (**d**) old wood treated with Au and Hap. Reused from an open access source [1].

**Figure 4 materials-13-02064-f004:**
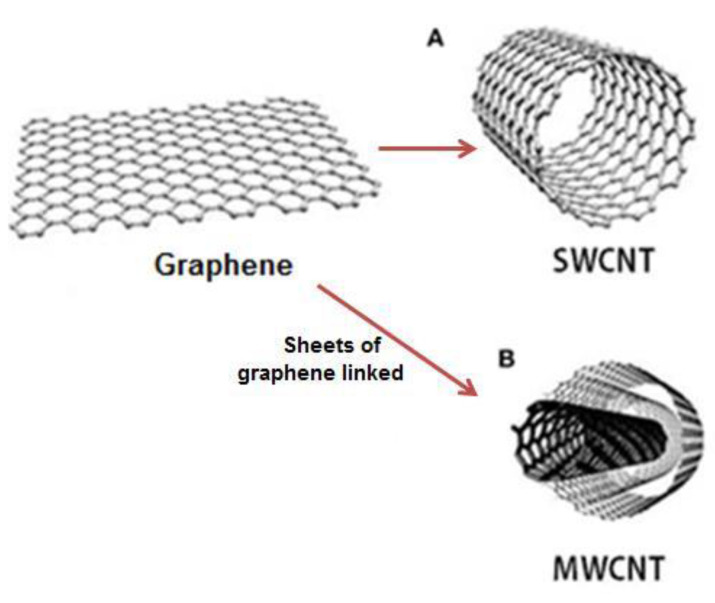
Graphene and carbon nanotubes as structures: (**A**) single-wall carbon nanotubes (SWCNT) and (**B**) multi-walled carbon nanotubes (MWCNT).

**Table 1 materials-13-02064-t001:** Nanoparticles used in the literature as possible enhancers for wood and their properties.

Nanomaterials	Properties	References
TiO_2_	UV protection, hydrophobicity, self-cleaning, fire resistance, protection against microorganisms, dimensional stability	[47,48,49]
ZnO	UV protection, resistance to fire and scratches, hydrophobicity, protection against microorganisms, dimensional stability	[50,51,52]
Au	Protection against microorganisms	[1,34]
Ag		[53]
MgO	UV protection, hydrophobicity, protection against microorganisms	[54]
FeO	UV protection, protection against microorganisms	[34]
SiO_2_	Fire resistance, self-cleaning, hydrophobicity, scratch resistance	[55]
CuO	Protection against microorganisms	[56]
HAp/Au	Anti-aging protection, mechanical resistance, hydrophobicity	[1]

**Table 2 materials-13-02064-t002:** Functionalized CNTs and their applications.

Structures	Applications	References
MWCNT-g-PCA-Au *	Lateral-flow immunochromatography assay for the screening	[181]
MWCNT-g-APC-Au	Optical nanosensor for determination of trace amounts of thiourea in spring water and orange peel	[182]
MWCNT-Au	Sensor for hydrogen peroxide determination	[183]
MWCNT-ms-Ag	Antimicrobial agent	[180]
SiO_2_-MWCNT/Ag	Water treatment (antimicrobial effect)	[184]
MWCNT-Ag	Antimicrobial agent	[185]
MWCNT-Ag	Plasmonic photo thermal therapy in melanoma cancer	[186]
MWCNT-TiO_2_	Photocatalytic activity	[187]
MWCNT-Ag/TiO_2_	Antimicrobial agent	[188]
MWCNT-ZnO	Bolometric sensor	[189]
MWCNT-ZnO/NiO	Textile dyes degradation	[190]
Cs-MWCNT/MgO	Antimicrobial agent	[191]
MWCNT-Fe_3_O_4_	Drug delivery system	[192]
MWCNT-Fe_3_O_4_	Magnetic sensor	[193]
SiO_2_-MWCNT	Nanodevice for medicine	[194]
MWCNT-CuO	Chemical sensors with high sensitivity or catalysts with high activity to organic volatiles at low temperature	[195]
MWCNT-CuO	Glucose sensor	[196]
CNT-Pt	Sensors for ozone gas detection	[178]

* PCA—poly(citric acid); Pt—platinum; ms—surfactant matrix; Cs—chitosan.

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
