# Peer review of "Nanomaterials Used in Conservation and Restoration of Cultural Heritage: An Up-to-Date Overview"

_materials, 2020, doi:10.3390/ma13092064_

Round 1

Reviewer 1 Report

GENERAL REMARKS

Text in lines 337-345 is not related to nanomaterials and should be removed.

There are many sentences in which Present and Past Tenses appear together what is not allowed in the English grammary, see lines: 51-52, 147-148, 379-380, 385-386, 568-569, 582-583, 583-584. 630-631, 635, 762-763, 850.

Sometimes instead of single form is plural and vice versa, e.g., „hydroxyapatite and its metallic derivatives seems” (line 69), also lines: 248-249, 498, 751.

DETAILED REMARKS

Line 14/15: degradation factors.; rather: degradation, factors cannot be a threat.

Line 16: as close as possible of their initial one.; sentence unclear, what means „initial one”?; besides – as close as possible to …

Line 73: figure; Figure

Line 80: Form this point of view,; From…

Line 97: which human discovered, remains the manufacture of paper; which has been discovered by humans…

Line 106: to provid; to provide

Line 152: nanoaterials; nanomaterials

Line 323-324: Comparing the antimicrobial activity of MgO nanoparticles on Gram positive and Gram

negative bacteria have been shown that this material is more effective; has been shown…

Line 349: fascinating properties; fascinating? Sounds strange

Lines 398-399: some of the most researched approaches; several more investigated approaches

Line 616: 16–55nm, 1.2µm; spaces required

Lines 628-629: that increasing the reaction temperature and the precursor gas flow the diameter of the MWCNTs increased,; unclear sentence

Line 669: Acid treatment, removes; comma to remove

Line 670: it can affects: …affect

Line 714: high-energy radiation were; … was

Line 750: Others; Other

Line 780: Van der forces Waals,; Van der Waals forces

Table 5: Hydrogen Peroxide Determination; why capital letters?

Line 837: An example of tubular materials is carbon nanotubes,; The carbon nanotubes are among examples of nanotubular materials

Summarizing: the paper is well-prepared, exhaustive, important for a certain group of readers, but it only needs of a one shortening and a number of removals of grammar errors.

Author Response

Dear Reviewer

  • The lines 337-345 were removed.
  • The English was corrected according to the English grammar.
  • The other detailed remarks were corrected according to your recommendations.

Thanks a lot for your suggestions.

Reviewer 2 Report

Dear authors,

It is my pleasure to assume the good quality and the excellent organization of your manuscript, titled " Nanomaterials Used in Conservation / Restoration of Cultural Heritage: An Up-to-Date Overview".

The text was written correctly, even though some mistakes have to be corrected. The ideas were well explained in the subsections and the conclusions were well done.

I presume it can be published in Materials journal after some minor revisions. Please, follow my comments and do the best to improve your work.

  1. Lines 41-43: the sentence should be changed since the English does not sound well

For example: "Currently, new systems (metallic oxides and hydroxides (TiO2, ZnO, Ca(OH), Mg(OH), Sr(OH)2 and metal nanoparticles (Au, Ag, Pt)) successfully tested on different artifacts and works of art have been reported in the literature [1, 6, 7]."

  1. Lines 53-56: the sentence must be supported by references.

Of note, other comments about words changes were made directly in the text.

  1. The authors must improve the scale shown in Fig. 3.
  2. I advise the authors to include the reference https://doi.org/10.1007/s10570-019-02861-8 in the subchapter of CNTs (carbon nanotubes). For example, line 689.
  3. Conclusions have two mistakes.

The reviewer,

Author Response

Dear Reviewer,

- The lines 41-43 were corrected.

- The scale of Figure 3 was improved.

- The reference https://doi.org/10.1007/s10570-019-02861-8 was included in chapter 4.

  • The mistakes from the conclusions were corrected.

Thanks a lot for your suggestions.

Reviewer 3 Report

Dear Editor,

I have read the manuscript “Nanomaterials used in Conservation/Restoration of Cultural Heritage: An Up-to-Date Overview”, submitted as a review to the journal Materials. I consider the topic well-chosen and interesting, and I think it makes sense to publish a review on the application of novel nanomaterials for improving the conservation of artifacts of cultural heritage. From this point of view I consider that such a work can deserve publication. Nevertheless, I don’t agree with the final result and I consider that a lot of work needs to be done in order to produce a final manuscript that can really serve as a proper review with added value to what is already available in the literature. I explain the reasons below.

Author’s present tubular materials, specially carbon nanotubes as promising nanomaterials for preserving artifacts of cultural importance. I think this is one of the interesting topics of the review, but also, is where my major concerns appear. During section 3, Authors describe the use of different metal and metal oxide nanoparticles in the conservation of different kind of artifacts. I think this section complies with what is expected for a review work: concise and brief explanations of the use of such nanomaterials and references to the works cited. Unfortunately, Authors fail to do the same in section 4.

During 6 pages (from page 10 to 16), Authors describe the structure of carbon nanotubes and the different obtention and purification methods. Next few pages (from 16 to 19) are used to describe the properties of the carbon nanotubes. If this is a review about the obtention methods and properties and applications of carbon nanotubes, Authors fail completely on citing the most relevant and pioneer works and also the most recent advances. Moreover, such a review makes totally no sense as there are many exhaustive examples of that topic in the literature, including many classical books. According to the title, Authors should review the use of CNTs on the conservation of cultural materials, but surprisingly they just relate CNTs to heritage conservation through five lines of text, with only three works cited. This is totally unacceptable if this is the hot topic of the review.

Therefore, from my point of view before this manuscript can be considered for publication, section 4 has to be totally rewritten. Only a few paragraphs must be used to describe CNTs properties and why they are interesting for conservation, and the most common obtention methods, citing only the most classical and relevant works. The rest of the section must be used to list the existing works in the literature that use tubular materials for heritage conservation, mainly CNTs, and to explain the results there in, similarly as Authors did in section 3 regarding the metal and metal oxide nanoparticles. If only three works of CNTs for conservation of cultural heritage can be found, then maybe it is too early for this review.

I also have detected some errors:

- Line 141: there is a typo error.

- Lines 195 and 197: I guess authors want to write sol instead of soil.

- Line 457 must be rewritten as it is confusing.

- Line 562: References 156 and 159 are the same.

- Line 741 and 751: When we refer to physical interactions between a solid and an adsorptive, the correct term is Adsorption instead of Absorption.

Author Response

Dear Reviewer,

Many thanks for your valuable comments and recommendation, which helped us to greatly improve our paper.

We made important language and style corrections, corrected the typos and tried to answer all your requests as good as possible. All changes were clearly revealed by track change.

We made important changes in chapter 4, the obtained methods of CNTs and purification treatments of these materials were shortened. It was kept only the properties of CNTs relevant for cultural heritage area. Chapter 4.4 was rewritten and new information was added regarding CNTs used in cultural heritage.

We corrected all the grammar and spelling errors, accordingly to your suggestions.

Thank you again for your suggestions.

Round 2

Reviewer 3 Report

I still consider that references cited to describe the production and functionalization of carbon nanotubes are far from being the most important ones. Nevertheless, Authors have rewritten the section 4.4 and now I consider that the review is worth of publication.